# Circulating Tumour Cells (CTC), Head and Neck Cancer and Radiotherapy; Future Perspectives

**DOI:** 10.3390/cancers11030367

**Published:** 2019-03-15

**Authors:** Vanathi Perumal, Tammy Corica, Arun M. Dharmarajan, Zhonghua Sun, Satvinder S. Dhaliwal, Crispin R. Dass, Joshua Dass

**Affiliations:** 1School of Pharmacy and Biomedical Sciences, Curtin University, Perth, WA 6102, Australia; vanathi.perumal@curtin.edu.au (V.P.); a.dharmarajan@curtin.edu.au (A.M.D.); crispin.dass@curtin.edu.au (C.R.D.); 2Radiation Oncology, Sir Charles Gairdner Hospital, Cancer Centre, Nedlands, Perth, WA 6009, Australia; Tammy.Corica@health.wa.gov.au; 3Stem Cell and Cancer Biology Laboratory, School of Pharmacy and Biomedical Sciences, Curtin Health Innovation Research Institute, Curtin University, Perth, WA 6102, Australia; 4Curtin Health Innovation Research Institute, Curtin University, Perth, WA 6102, Australia; 5Discipline of Medical Radiation Sciences, School of Molecular and Life Sciences, Faculty of Science and Engineering, Curtin University, Perth, WA 6102, Australia; Z.Sun@exchange.curtin.edu.au; 6School of Public Health, Faculty of Health Sciences, Curtin University, Perth, WA 6102, Australia; S.Dhaliwal@curtin.edu.au

**Keywords:** circulating tumour cells, circulating cancer stem cells, radiotherapy, ctDNA, cf DNA

## Abstract

Head and neck cancer is the seventh most common cancer in Australia and globally. Despite the current improved treatment modalities, there is still up to 50–60% local regional recurrence and or distant metastasis. High-resolution medical imaging technologies such as PET/CT and MRI do not currently detect the early spread of tumour cells, thus limiting the potential for effective minimal residual detection and early diagnosis. Circulating tumour cells (CTCs) are a rare subset of cells that escape from the primary tumour and enter into the bloodstream to form metastatic deposits or even re-establish themselves in the primary site of the cancer. These cells are more aggressive and accumulate gene alterations by somatic mutations that are the same or even greater than the primary tumour because of additional features acquired in the circulation. The potential application of CTC in clinical use is to acquire a liquid biopsy, by taking a reliable minimally invasive venous blood sample, for cell genotyping during radiotherapy treatment to monitor the decline in CTC detectability, and mutational changes in response to radiation resistance and radiation sensitivity. Currently, very little has been published on radiation therapy, CTC, and circulating cancer stem cells (CCSCs). The prognostic value of CTC in cancer management and personalised medicine for head and neck cancer radiotherapy patients requires a deeper understanding at the cellular level, along with other advanced technologies. With this goal, this review summarises the current research of head and neck cancer CTC, CCSC and the molecular targets for personalised radiotherapy response.

## 1. Introduction

The worldwide incidence of head and neck cancer is more than 600,000 cases with 350,000 deaths each year [1]. In Australia, it is expected to rise to about 5061 new cases in 2018, including 3725 males and 1336 females, compared to 4409 cases in 2013 [2,3]. Some of the associated confounding factors include tobacco-chewing, smoking, alcoholism, poor oral hygiene and p16 (cyclin-dependent kinase inhibitor 2A, multiple tumour suppressor 1) status in oral cancers. Typically, there are five main types of head and neck cancer: laryngeal and hypo pharyngeal (voice box), nasal cavity and paranasal sinus (behind the nose), nasopharyngeal in the upper part of the throat (behind the nose), oral and oropharyngeal (mouth, tongue and salivary glands) [4,5,6,7,8,9,10]. These tumours predominantly originate from the squamous cells lining the surfaces of mouth, nose and the throat. The majority of head and neck cancers are squamous cell carcinomas (HNSCC). Despite recent improvements in loco-regional control, 50–60% of HNSCCs develop loco-regional recurrence, a further 20% progressing to distant metastasis and therefore treatment failure [11]. Hence, globally the diagnosis and prognosis of HNSCC remains a challenge [12].

These statistics indicate that there is an immediate need for improved therapy modalities specifically for the HNSCC patients who are at the risk of loco regional or distant metastasis. In clinical practice, it may be difficult to obtain tumour tissue from patients for gene alteration discoveries to tailor treatment. Currently, radiotherapy alone or in combination with chemo-radiotherapy has been reasonably effective for HNSCC but there is room for improvement [13,14,15]. Hence, the combined effort of researchers and clinical investigators will expand the horizons in discovering new effective biomarkers for clinical utility [16,17].

Despite the emergence of recent state-of-the-art radiotherapy modalities such as Image-Guided Radiation Therapy (IGRT), Intensity-Modulated Radiation Therapy (IMRT), Volumetric Modulated Radiation Therapy (VMRT) or Stereotactic Ablative Body Radiotherapy (SABR), there is a limitation on the precise dose delivery associated with tumour volume and on the biological effect [18] in determining the radioresistance and sensitivity index of the patient. Radioresistance and radiosensitivity may vary depending on the cell type and origin and the genetic makeup of the patient. Cancer stem cells (CSCs) are more resistant to radiotherapy [19,20]. Failure in repairing the double strand breaks of DNA by radiotherapy accumulates mutation, causing genomic instability [21,22]. Currently, radiation oncology is being revolutionised into a new era with more precise and exciting radiobiological advancement technologies by using CTCs and CCSCs. Ionising radiation to the primary tumour target can affect the non-primary tumours favourably or unfavourably, which is termed an “abscopal” effect. From an oncologist’s point of view, reduction in the tumour size is the measured criteria, whereas from a biologist’s point of view, the measured criterion is the epigenetic modification causing tumorigenic alteration and cell death in healthy tissues. The use of high-dose radiotherapy fractionation in combination with appropriately improved systemic agents will be more beneficial in controlling the tumour burden and in protecting healthy tissues from local recurrence [23]. Thus, the rationale for identifying active cancer by investigating the presence of CTCs is not only a diagnostic tool but will also serve as a proof of concept in guiding the efficacy of current clinical radiotherapy treatment modalities [24]. 

## 2. CTC Origin and Methodologies

In 1869, the Australian physician Thomas Ashworth [25] discovered the existence of CTCs in blood during the autopsy of a metastatic cancer patient. Later on, in 1889, Paget [26] postulated the “seed and soil theory” based on his observations in metastatic breast cancer. Now CTC isolation and detection have progressed into a new era. Currently, there are different approaches available for the identification of CTCs based on the physical properties, expression of biomarkers or functional characteristics of the tumour [27,28].

Solid tumour CTC isolation is commonly based on the biological properties which are enriched either by EpCAM (epithelial cell adhesion molecule) positive selection or CD2, CD14, CD16, CD19, CD45, CD61, CD66b and Glycophorin A negative selection of cell surface antibodies (such as Stem Cell Technologies Human CTC Enrichment Kit, Rosette sep product CTC kit, MACS cell separation-Miltenyl Biotec Immunomagnetic bead kit) [3,29,30,31,32]. CTC isolation can also be enriched by ficoll density gradient, deformability or electric charges dielectrophoresis (DEP), flow cytometry, immuno-microbubbles and protein epithelial immunospot (EPISPOT) methods. The other microfluidic platform system (CTC-chip) [32] and isolation via size of epithelial tumour cells (ISET) filtration methodologies are based on the size of the CTC specific to tumour type. For further precise validation and to avoid false negative findings, immunostaining or reverse transcription PCR (RT-PCR) methods can be supported by a genomic approach using fluorescent in situ hybridisation (FISH) or single cell analysis. Among various CTC methodologies, CellSearch^®^ System (Veridex, Huntington Valley, PA 19006, USA) CTC test is the only method that has been approved by the FDA (Food and Drug Administration) such that it can be used as a surrogate marker of overall and progression-free survival for only few types of cancer like breast, prostate and colorectal cancers [33]. In a comparative study of CTC methodology on breast cancer, Kallergi et al. [34] reported ISET platform performance for the best recovery than the CellSearch system. Morris et al. identified hepatocellular CTCs in 19 out of 19 (100%) patients by ISET methodology and 14 out of 50 (28%) patients by Cell Search system suggesting the accuracy in sensitivity and specificity [35]. Khoja et al. [36] compared the detection of CTC in 54 pancreatic cancer patients with ISET and CellSearch system. The percentage of detection was higher; 93% by ISET and 40% by CellSearch. Pailler et al. [37] also concluded ISET methodology to be 100% sensitive compared to 33% with CellSearch in non-small cell lung cancer (NSCLC) patients. Another study conducted by Krebs et al. [38] explains the advantages and disadvantages of using both CellSearch and ISET approaches for a complementary role. CTC recovery rate with the ISET method was 95% and 80% compared to 52% and 23% with CellSearch; however, the results obtained were only from small patient groups of 10 and 40, respectively. In a direct comparison study of CellSearch and ISET, Farace et al. [39] discussed the limitation of using the EpCAM antigen (CellSearch assay) and cell size (ISET assay) in 20-patient cohorts for metastatic breast, prostate and non-small cell lung cancer. They concluded the ISET methodology to be a more accurate clinical tool for breast, prostate and non-small cell lung cancers. Hofman et al. [40] also compared the efficacy of CellSearch assay^TM^ and the ISET methods in 210 patients undergoing radical surgery for non-small-cell lung carcinoma (NSCLC). They concluded that both methods were good prognostic markers for CTC detection in this patient group. CTC detection methods utilised in studies investigating head and neck cancers are summarised in Table 1. Due to their reported sensitivity and specificity, ISET and Maintrac methodology have been widely used for clinical utility at the NutriPATH, National Institute of Integrative Medicine (NIIM) and Genostics in Australia [41,42,43].

## 3. Radiotherapy Response on Circulating Tumour Cells, ctDNA and cfDNA 

To date, there have only been a few published studies investigating the impact of radiation therapy on CTCs [24,51,61,63,69,74,75]. In HNSCC, Buglione et al. [61] investigated the role of CTC in patients receiving radiotherapy alone; however, the results were grouped with patients who had also received chemotherapy and the sensitivity was low (30%) [61]. Radiotherapy in combination with chemotherapy has shown reduction in HNSCC and prostate CTC [51,61,65,70,75,76,77,78]. Research indicates radiotherapy makes the tumour cells more aggressive and potentiates the circulation of non-small lung cancer (NSCLC) cells [24]. The successful use of conventional fractionated high-dose radiotherapy (~2Gy per fraction for 30 days) can result in the destruction of clonogenic tumour cells through DNA damage and cell cycle arrest [24], through to mitotic catastrophe, apoptosis, necrosis, or autophagy [79,80,81], depending on the dose. Cellular radioresistance and radiosensitivity depends on cell type and origin, cell cycle, and the genetic background. Martin et al. [24] indicated an increase in CTC counts during early phase radiotherapy. Dorsey et al. [74] measured human telomerase reverse transcriptase (hTERT) activity specific to cancer cells as a threshold for live CTC positivity in NSCLC patients receiving radiotherapy. Lowes et al. [75] also reported that CTC could be used as a predictive biomarker in clinical decision making for radiotherapy favourable prostate cancer patient group. Our research group [14,15] reported the significance of favourable hyperbaric oxygen therapy for chronic radiation-induced tissue injuries. Administering low-dose cisplatin, compared to a high-dose treatment regimen, was well tolerated when used in combination with radiotherapy in head and neck cancer patients. Both of the studies in Figure 1 and Figure 2 comprise the basis of the research expertise and the scope for assessing the influence of CTCs when using radiotherapy and chemoradiotherapy. 

More recently, blood-based prognostic and predictive biomarkers for radiotherapy have drawn much attention and excitement in radiation oncology. In this section, we reviewed and explored all the noted uses of HNSCC circulating tumour DNA (ctDNA) for radioresistance and radiosensitivity diagnostic approaches. 

Primary tumour information and metastases at different sites within the same patient may have different genomic characteristics due to tumour heterogeneity [82]. Therefore, tumour biopsy alone, which is currently used in clinical practice, may not reveal sufficient information for radiotherapy response decision-making. Combining ctDNA and cell-free DNA (cfDNA) genetic information in radiation oncology can complement the current available technologies for future radiosensitising therapeutic intervention. However, evidence has shown that there is a radiation risk from CT scans inducing malignancy, suggesting that it may not be an ideal method for early cancer detection [83].

cfDNA is unstable and short-lived in the bloodstream [84,85,86,87], having a mix of malignant and non-malignant cell properties. It is released into the circulation through apoptotic and necrotic cell death mechanisms [85,88,89,90]. The DNA are short fragments of 166 bp sizes in low concentration. Increase in cfDNA level suggests the presence of residual tumour cells in plasma [91]. Interestingly, both mitochondrial DNA (mtDNA) and genomic DNA (gDNA) constitutes the total cfDNA [90]. The circulation of nucleosome, mutations, methylation, chromatin modifications, and virial DNA, like human papillomavirus (HPV), hepatitis B virus (HBV), and Epstein–Barr virus (EBV) in cfDNA has the potential to monitor the anticancer therapy response in nasopharyngeal cancer and head and neck cancer [92,93,94,95,96,97,98,99,100]. In some clinical studies, the increase in cfDNA fragments in blood plasma and serum has been associated with poor prognosis [90].

Protein-based circulating biomarkers are specific to one cancer type, and it is not feasible to compare and generalise this to all cancer types [101]. On the other hand, ctDNA and cfDNA [17,101,102] in all cancer types accumulate somatic mutations, which can be correlated with any cancer type treatment regimens [17,101]. In metastatic breast cancer patients, Dawson et al. [87] measured the ctDNA mutant allele fraction and identified progression 5 months before imaging, suggesting that ctDNA quantification is more beneficial than imaging [87]. Other authors have suggested that ctDNA should be considered for clinical use in post-treatment follow-up of early stage cancer patients in whom there is no or minimal residual disease, given the higher sensitivity of such assays [103,104]. 

Next-generation sequencing technology platforms for ctDNA analysis was developed 20 years ago [105,106,107,108] as either targeted or untargeted approaches. Targeted approaches include BEAMing Safe-SeqS, TamSeq, and digital PCR, RTq PCR [109] to detect single-nucleotide mutations for limited genome regions, focusing on predefined gene hot spots like BRAF, KRAS and EGFR. The untargeted approach focuses on whole exome sequencing (WES) and whole genome sequencing (WGS) for screening genomic aberration by hypomethylation [110] insertions or deletions, acquired by therapy resistance. Murtaza et al. [111], using exome-wide analysis, quantified very high ctDNA levels for 1 to 2 years in 6 cancer patients and also reported on several mutations confirming drug resistance acquired post-treatment [111]. More recently, a novel method called cancer personalized profiling by deep sequencing (CAPP-seq) has demonstrated the advantage of combining the untargeted and targeted approaches, which can then be generalised to all types of cancers with high specificity [105]. The major advantage of ctDNA application in radiation oncology is studying the tumour kinetics for radiotherapy response assessment. Until now, using CAPP-Seq has differentiated the normal and residual disease ctDNA changes in NSCLC patients treated with fractionated and stereotactic ablative radiotherapy [105]. Alternatively, ctDNA usage can be applied to patients developing symptoms or perform imaging when there is a rise in the ct DNA levels. 

Measuring Circulating Epstein Barr virus (EBV) DNA and human papilloma virus (HPV) DNA in HNSCC patients can be used as a biomarker for before and after radiotherapy and chemo-radiotherapy response. Successful radiotherapy treatment should decrease the level of circulating EBV and HPV DNA [97,98,100,106,107,108,112]. In a different paradigm, where ctDNA levels remained stable and post-treatment imaging revealed complete response for radiation therapy, months later, a patient developed metastases in multiple organs, suggesting ctDNA level was an indicator of micrometastatic disease progression [101]. Lo et al. [98] observed that plasma EBV DNA increased in the first week of radiation therapy and subsequently decreased in nasopharyngeal carcinoma patients. Thus, ctDNA kinetics as a biomarker could help clinicians deliver tailored radiotherapy and/or adjuvant systemic therapy for individualised treatment rather than following a one-size-fits-all approach [98,99]. 

Currently, only clinical radiotherapy imaging data, as indicated in Figure 1A,B and Figure 2A,B, guide clinicians in treatment response when using radiotherapy alone in HNSCC patients. Towards precision medicine in radiation oncology, CTC counts, ctDNA and cfDNA, along with medical imaging modalities, will play a beneficial role in predicting patient prognosis. 

## 4. CTC Lines, CCSC Derived CSC Organoids for Future Tailored Radiotherapy 

CTC culturing for functional studies from cancer patients is logistically challenging. Currently, for understanding CTC, CCSC biology and tumour heterogeneity, all race cell lines or preclinical models for developing new efficient novel drugs are not commercially available [113]. Functional analysis involves understanding the viable CTCs metastatic-initiating potential using in vitro and in vivo models [114,115,116,117,118]. Based on the existing HNSCC chemo-radiotherapy literature, it is expected that no isolated CTCs using different methodologies post-treatment can be cultured in vitro. Cayrefourcq et al. [115] established the first CTC permanent cell line from the blood sample of a colon cancer patient and validated the tumour-forming capacity in SCID mice [115]. Interestingly, it shared similar characteristics to those of the primary tumour, possessing epithelial stem-like features. In a heterogeneous population of cells in circulation, very few, depending on the genetic makeup, can be cultured successfully for real-time assessment of radiotherapy treatment. Until now, very few studies have validated HNSCC CTC expansion and culturing as tumour spheres [3,11,50,119]. 

Cancer stem cells (CSCs) are cells possessing self-renewal properties with high metastatic potential and resistance to therapies. The major reason for treatment failure in anticancer therapy modalities is due to the presence of CSCs. CTCs expressing stem cell markers in the circulation are known as circulating cancer stem cells (CCSCs) and are highly tumorigenic compared to tissue CSCs [120]. CSCs and non-CSCs possess common invasive/migratory features such as epithelial-mesenchymal transition. Regarding CCSC, Patel S et al. [29] explored the role of CD44v6 and Nanog stem cell markers in oral squamous cell carcinoma population. Commonly used CSC markers like CD133, CD44 and ALDH were associated with poor patient outcome. Among them, CD44 is highly expressed in HNSCC cancer stem cells and circulating cancer stem cells. The over-expression of CD44+ and CD133+ cells provides resistance in tumours and the capability for disease relapse after chemo/radiotherapy in immunodeficient mice [121]. Therefore, selective targeting of these markers should be utilised to deliver cytotoxic drugs to CSCs, as one Wnt pathway antagonist (sFRP4) in HNSCC CSCs decreased CD44, ALDH, ABCG2 and ABCC4 drug-resistance expression [122]. Furthermore, detailed head and neck cancer CSC biomarkers and their transcription factors are reviewed as CD44, CD133, ALDH, cMET and Oct-4, Nanog, Sox2, respectively [123,124].

With regard to HNSCC HPV status, HPV-positive patients responded to radiation treatment better when compared to HPV-negative patients. Vlashi et al. [125] tested the CSC hypothesis to inherent radiosensitivity and radioresistance in HPV-positive and -negative HNSCC cell lines. HNSCC HPV-positive and -negative status CSC demonstrated radiation effects that were similar to HNSCC HPV-positive and -negative cell lines results. Additionally, radiotherapy induced the dedifferentiation of head and neck cancer cells into stem cells and increased plasticity by Yamanaka reprogramming factors (transcription factors Oct4 (Pou5f1), Sox2, cMyc and Klf4) in HPV-negative cell lines [125].

In summarising the literature, the CCSC findings from several studies reveal that CTCs from head and neck, lung, breast and prostate cancer can be cultured as spheroids or organoids for more than 6 months, and these CTCs possess self-renewal cancer stem cell properties and exhibit a radioresistant and chemoresistant phenotype with a high metastatic potential [109,120,126,127,128,129]. Recently, binding sites for OCT4, NANOG, SOX2 and SIN3A in breast cancer CTC clusters were shown to be hypomethylated to what is seen in embryonic stem cell biology [110]. 

The authors suggested the CD44^+^ CSC-like cell population possessed dose-dependent resistance to cisplatin in the peripheral blood circulation as do CCSC [29]. The same group again reported the anticancer effect of curcumin inhibiting the stem cell features (self-renewal potential) of non-small cell lung carcinoma CCSC [130]. Recently, another group, Zhang S et al., compared the metastatic and invasiveness characteristics of circulating gastric cancer stem cells (CGCSCs) and tissue gastric cancer stem cells (TGCSCs) in human gastric adenocarcinoma patients [131]. Interestingly, Chen et al. [132] reported that CGCSCs showed greater aggressive phenotype to TGCSCs when isolating CGCSCs and TGCSCs from blood and tissue of gastric adenocarcinoma patients. They also successfully developed in vitro cell culture expansion with the same phenotype for several passages [132]. In a correlation study of CTC and CCSC from non-metastatic breast cancer patients, CTC was considered to be more appropriate for therapy assessment and CCSC as an independent prognostic marker for treatment failure and tumour recurrence [133]. Similar findings were reported for use of CD133 and CD44v9 CSC markers for predicting recurrence, prognosis, and treatment efficacy in colorectal cancer patient CTCs [134,135,136].

Kantara C et al. also developed a novel diagnostic assay, expressing CSC markers (DCLK1/LGR5) with CD44/Annexin A2 for detecting colon cancer CCSCs in the blood of colonic adenocarcinomas patients [137]. Similarly, Grillet F, et al. [138] developed a colorectal cancer (CRC)-expressing cancer stem cell phenotype for drug testing and monitored the metastatic potential in the real world for clinical practice. The outcome of the study is now in clinical trial registration (ClinicalTrial.gov NCT01577511) for individualised treatment [138]. Interestingly, a patent-pending study on doxorubicin and aspirin as low dose combination therapy in patient-derived CTC clusters reduced CD 44 and 24 CSCs and tumour relapse [139].

## 5. Head and Neck Cancer Treatment Modalities and Application

Management of head and neck cancers to date has relied on the conventional treatments of surgery, radiotherapy and chemotherapy. Initiating treatment of this group of cancers by a molecular approach is innovative, and currently there is limited evidence available regarding the effects of CTC concentration during radiotherapy treatment to gauge responses and modify radiation doses. The CTC genes regulating the radiotherapy kinetics (including dose rate and total dose) have not been identified to date with particular applications of radioresistance and radiosensitivity. Furthermore, CTC gene expression changes after radiotherapy, and chemotherapy management is still in the infancy stages of discovery. At present, much of the work done has been focused on signalling pathways. 

Deregulated signalling pathways contribute to oncogenic cellular transformation and resistance. Several gene therapy strategies with different vector delivery mechanisms are being investigated in clinical trials for HNSCC in the UK [140,141,142,143]. Radiotherapy with gene therapy will be a novel approach to treat HNSCC in the future [144,145]. Gene expression profiling and identification of biomarkers will further complement the current major clinicopathologic challenges, including staging, lymph node, and distant metastasis, treatment choice and patient outcome. In 2007, the Netherlands Cancer Institute (NKI), using NKI array method determined the gene expression profile as a good prognostic value for chemo-radiotherapy (cisplatin-based) in Phase II and randomized Phase III trials of advanced HNSCC patient group [146]. Later, 10 hub genes (AR, c-Jun, STAT1, PKC-β, RelA, cABL, SUMO1, PAK2, HDAC1 and IRF1) were developed using a system biology-based gene expression classifier for regulating radio-sensitivity genomic adjusted radiation dose (GARD) [147], including the radiation response of DNA damage, histone deacetylation, cell cycle regulation, apoptosis and proliferation [148,149,150].

In a collaborative study in 2009 between the Moffitt Cancer Centre, Research Institute USA and the Netherlands Cancer Institute, the multigene expression model for predicting the intrinsic tumour radiosensitivity and treatment response in patients was developed [151]. Subsequently, Torres-Roca et al. measured the GARD values [147] in 8271 primary tissue samples consisting of 20 different disease sites. The GARD values ranged from (1.66 to 172.4). Gliomas and sarcomas resulted in lower values and cervical and oropharyngeal HNSCCs resulted in higher values. The GARD approach emphasises only the radiosensitivity parameters with the potential for further optimising the current radiation dose, using tumour-specific genomic data as a novel genomic radiotherapy prescription framework [152]. Conventional HNSCC radiotherapy treatment guidelines currently follow a ‘one size fits all’ modality, guided by the TNM staging system, which is currently under review. The new 8th Edition TNM staging takes into account oropharyngeal p16 positive cancer, which is down-staged accordingly to move away from the ‘one size fits all approach’. Along with precision medicine tools such as genomics, radiomics and mathematical modelling, the GARD approach could open new doors in reducing toxicity and improving patient outcome. Supplementing the GARD approach with information gained by CTC genomics could further guide personalized treatment by tailoring radiotherapy dose modifications to individual patients. Understanding the role of EGFR in HNSCC metastasis and treatment efficacy is essential for improving treatment outcomes [153,154,155,156,157]. Often, patients with drug resistance EGFR mutations are limited to treatment with tyrosine kinase inhibitors to target the epidermal growth factor receptor gene (EGFR) [154]. Monitoring HNSCC CTCs and their expression of EGFR and pEGFR in the blood has been implicated as a good predictor for radio-chemo therapy response [55,57]. Studies have shown that CTC concentration increased in the presence of radiotherapy in stage IVA/B, HNSCC patients. However, addition of cetuximab decreased the CTC number compared to cisplatin/5-fluorouracil. The treatment was also effective in decreasing the pEGFR expression [158]. Overall, EGFR status can be used as a chemo-radiotherapy and radiotherapy indicator along with other CTC epithelial markers like EpCAM and cytokeratin. Evidence also proves that EGFR plays a vital role in maintaining the HNSCC CTC clusters (possessing the stem cell features) showing metastatic potential. Targeting EGFR in these clusters might lead to a more precise treatment approach. 

Gemmill et al. and Wagner et al. [159,160] critically reviewed the novel usage of Cetuximab for HNSCC in a large randomised Phase III clinical trial. Cetuximab in combination with radiotherapy showed good loco-regional control compared to radiotherapy alone with less acute toxicity by inhibiting epidermal growth factor receptor (EGFR) [159,160,161]. Later, Tinhofer et al. showed a decrease in CTC number in combination with radiotherapy [158]. Cisplatin (cis-diammine-dichloro-platinum^II^) is a cytotoxic drug currently used in the treatment of several cancers and is effective through causing DNA damage or apoptosis of tumour cells. The usage of cisplatin as a radiation sensitiser in the treatment of advanced HNSCCs are summarised in phase II clinical trials [162,163,164,165,166,167,168]. From an Australian radiotherapy perspective, our own research group conducted a retrospective study that showed the efficacy and tolerability of cisplatin in HNSCC patients receiving radiotherapy treatment. [15]. An ongoing international randomised controlled trial being run by the Trans-Tasman Radiation Oncology Group (TROG 12.01—HPV Oropharynx) is comparing the side-effects of cetuximab versus cisplatin, as they both possess different antitumor effects on radiation therapy. In an American randomised phase III trial (RTOG 0522), the outcome did not improve when cetuximab was combined with cisplatin and delivered concurrent with radiotherapy [169]. Similar studies utilising the hypoxic cytotoxin tirapazamine combined with cisplatin and radiation have been performed, including the TROG 02.02 Phase III “HeadSTART” trial, which did not find any significant improvement in outcome for head and neck cancer patients [170]. In a Phase III Danish Head and Neck Cancer Study (DAHANCA), hypoxic radiosensitizer Nimorazole (Nimoral) was found to significantly improve the hypoxic radio-sensitisation of supraglottic laryngeal and pharyngeal carcinoma with minimal side effects [171]. Nimorazole is currently being investigated in the EORTC 1219 ROG-HNCG TROG 14.03 trial using the 15 hypoxic gene signature in HPV/p16-negative SCC patients to identify hypoxic cell signature for radiotherapy treatment outcomes. 

## 6. Targeting CTC by Wnt Signalling for Radio Ligand-Based Therapies

Wnt signalling plays a significant role in radio resistance. To our knowledge, the role of radio resistance and the Wnt signalling mechanism is not yet elucidated in any of the cancer circulating tumour cells and its associated circulating cancer stem cells. There is, however, limited literature regarding Wnt signalling in circulating tumour cells. Some data exist surrounding the non-canonical CTC gene signatures and WNT2 candidate gene in pancreatic CTC, Wnt expression contributing to antiandrogen resistance by single cell analysis RNA-seq in prostate CTCs [172,173,174].

Since 1997, our research group has extensively published on Wnt signalling and the role of secreted frizzled related protein (sFRP4) and its associated domains, cysteine rich domain (CRD) and netrin-like domain (NLD) as an antiangiogenic protein in various cancers such as brain, breast, ovary, prostate, mesothelioma, including HNSCC CSCs and now recently in reproductive CSCs of breast, prostate and ovary [122,175,176,177,178,179,180,181,182,183,184,185,186,187]. Jun et al. [188] explored the molecular mechanism of Wnt signalling-induced radioresistance and DNA double-strand break repair by LIG4-DNA ligase through β-catenin in colorectal cancer. 

Based on our published data, we hypothesise the sFRP4 and its associated domains CRD and NLD will modulate the WNT 3a gene expression-induced radioresistance in HNSCC CTC and CCSCs and could reverse the effect of radioresistance. By diligently moving forward one step at a time, this potent and selective inhibitor may assist drug discovery and has potential to reverse the effect of radioresistance by targeting human HNSCC-derived CTCs and CCSCs resulting in a novel radiotherapy breakthrough for anticancer therapeutic modalities in the clinic [122,176,183,184,188]. 

From a clinical perspective, a high level of nuclear β-catenin accumulation was detected in colorectal cancer, HNSCC [189,190], ionizing radiation-induced glioblastoma [191] and in CTCs of prostate and breast cancer and myeloid leukaemia. Targeting β-catenin is also a promising therapeutic approach in overcoming radioresistance [192,193,194]. Interestingly, with respect to the CSC hypothesis, a Wnt signalling pathway member leucine-rich repeat-containing G-protein coupled receptor 5 (LgR5) showed significance in oesophageal adenocarcinomas (EAC) with and without Barrett’s Oesophagus (BE) [195]. 

c-Met is frequently altered in 90% of HNSCC tumours by mutation and overexpression contributing to HGF/cMet signalling [196,197,198,199]. Several anticancer tyrosine kinase inhibitors and the ongoing clinical trials targeting HGF and c-Met receptor are summarised by Rothenberger NJ et al. [196] Cirzotinib (PF-2341066) is a tyrosine kinase inhibitor that when used in combination with docetaxel and cisplatin showed synergistic antitumor effects in HNSCC CSCs in a patient-derived xenograft (PDX) model [197,199]. Previous findings show crosstalk between inhibiting cMET and WNT signalling. Furthermore, as proof of concept Sun et al. reported on the role of c-Met/FZD 8 in inhibition of patient derived CSC-like cells by selective c-MET inhibitor PF-2341066 [197]. Thus, use of c-MET inhibitors for antagonising Wnt signalling in HNSCC CSCs is warranted. Another interesting finding which may be also relevant to HNSCC CTC detection is that highly sensitive (80%) c-MET FISH-based platforms have been seen in gastric, pancreatic, colorectal, bladder, renal, and prostate cancers [198]. Studies have shown that few EGFR inhibitor-positive HNSCC patients treated with routine cetuximab also acquired resistance to c-MET [200,201]. Developing β-catenin and c-MET-based techniques will also contribute to enhancing the sensitivity of the current established CTC detection methods and aim to improve the survival of HNSCC patients. The overall prospects of HNSCC CTCs research in radiation oncology and progressing towards unexplored CTC-cancer research are summarised in Figure 3 and Figure 4. 

## 7. Conclusions and Future Directions 

This review article provides mounting evidence and scope for using CTC counts, CTC-derived CCSC and ctDNA as a prognostic and predictive marker for current head and neck individualised radiotherapy and chemo-radiotherapy response. Tracking CTC counts during radiotherapy treatment will measure the outcome and disease resistance leading to recurrence especially in loco regionally advanced disease with a significant potential for clinical application. In future, it may also set the stage for developing CTC radiosensitive index by GARD in the clinical setting for tailored radiotherapy dose prescription clinical practice. Today cost effective NGS and other sequencing platforms can also guide the radiotherapy genomic alterations response. Our group is currently developing CTC, circulating cancer stem cells and imaging guided genetic information (model) framework for radiotherapy resistance and sensitivity in HNSCC for clinical practice. We are confident that the proposed research will also contribute to the development of new therapeutic strategies for radiotherapy precision medicine. Identification of radiotherapy/chemo-radiotherapy relapsed HNSCC CTC and CCSC subpopulation and associated genetic material will be a valuable clinical tool to improve the current treatment approach for this disease. Developing a less invasive CTC, CCSC as a prognostic and predictive marker will be of added advantage in clinical practice for sequential radiotherapy and chemo-radiotherapy assessment. Furthermore, investigating WNTs, c-Met, PI3K, AKT/mTOR, CCSC epithelial markers and stem cell markers in radiotherapy HNSCC CTCs has the potential to generate future radio-sensitisation modalities. Globally, current drug discovery is an expensive and time-consuming process with high failure rates in patients. Future studies investigating CTCs on early and advanced cancer models offer a less invasive approach compared to tissue biopsies and can facilitate further understanding of tumour heterogeneity. The aim is to tailor treatment to individual patient requirements based on information gained by conducting chemo-radiation testing on their own CTC/CCSC organoids. This would then identify the most suitable FDA-approved chemo-radiation regimen for their particular cancer. Using the advanced CTC/CCSC organoid models, resistance and sensitivity rate can be predicted for therapy response prior to administering the therapy to the patients, avoiding unwanted toxicity burden. Patient CTC organoids approach in the real world is faster and less expensive with more control than animal models. Our goal is to introduce CTC/CCSC as companion diagnostics and clinical intervention for current available treatment prediction. Identifying and targeting specifically the CSC in CTC using surface markers and combining with radiotherapy will be major advancements in the field of cancer research. Introducing such robust platform discoveries and translating it to human research in radiotherapy treatment will help resolve the current challenges of potentially incurable cancer recurrence.

## Figures and Tables

**Figure 1 cancers-11-00367-f001:**
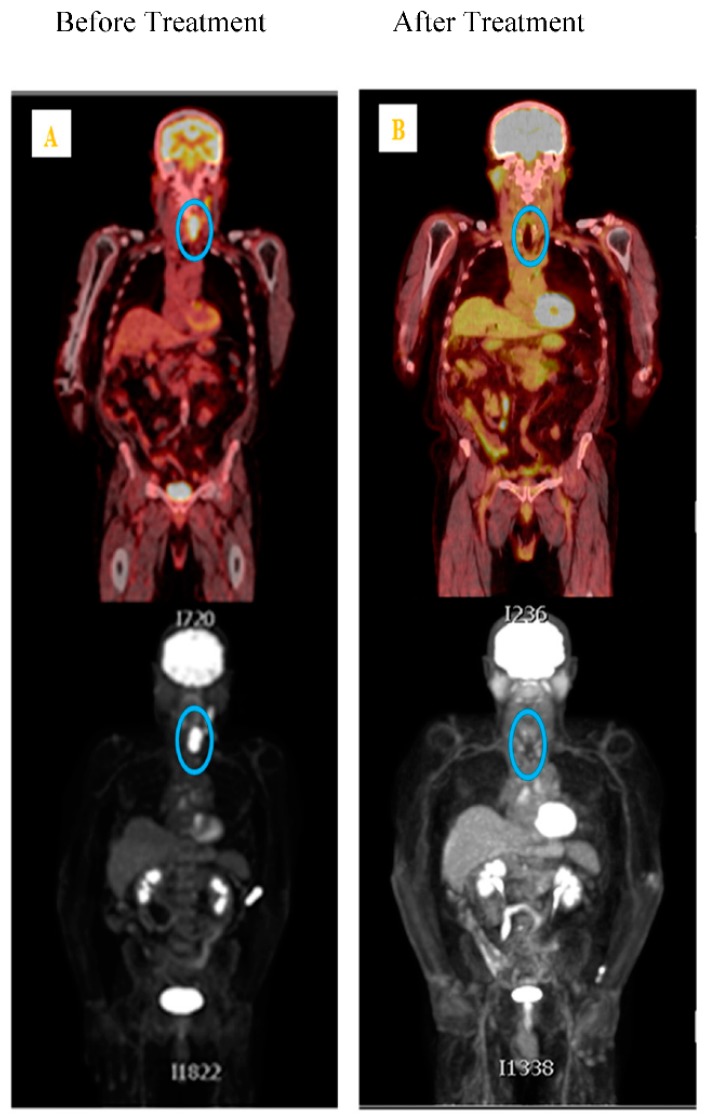
PET scan of Male pyriform fossa squamous cell carcinoma stage T4aN2bM0 patient without surgery, treated with radical radiotherapy delivery 70Gy/35# over 7 weeks followed by cisplatin 100 mg/msq given on week 1, week 4 and week 7 of radiotherapy. (**A**) Left panel blue circle showing the tumour spot before treatment on the 23 October 2014; (**B**) Right blue circle panel showing the antitumor effect of radical radio chemotherapy after treatment, showing no regional palpable lymphadenopathy, oral capsule was clear, showing no local recurrence, patient follow up after two years on 11 May 2017.

**Figure 2 cancers-11-00367-f002:**
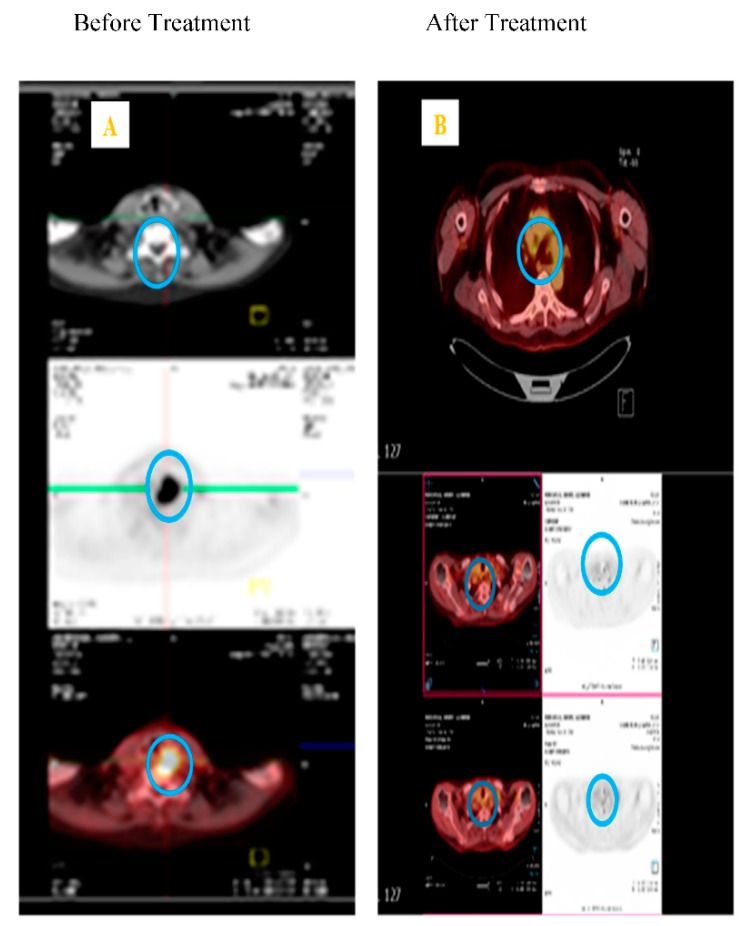
CT scans of the neck region in pyriform fossa squamous cell carcinoma patient showing the tumour area and no local recurrence. (**A**) Right panel blue circle showing clear palpable lymphadenopathy. (**B**) Left panel blue circle showing the tumour regression after two years, nasendoscopy also showed no local recurrence.

**Figure 3 cancers-11-00367-f003:**
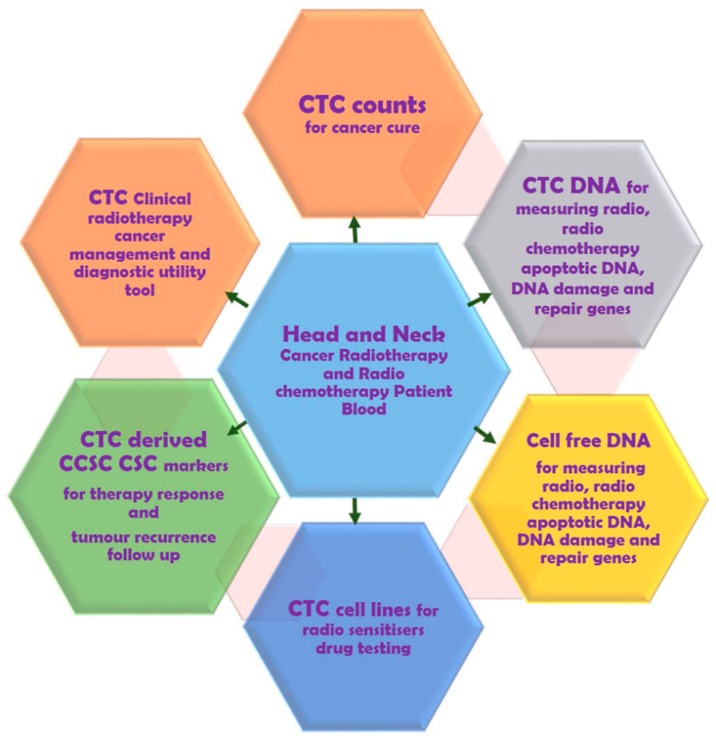
Potential of head and neck circulating tumour cell research in radiation oncology.

**Figure 4 cancers-11-00367-f004:**
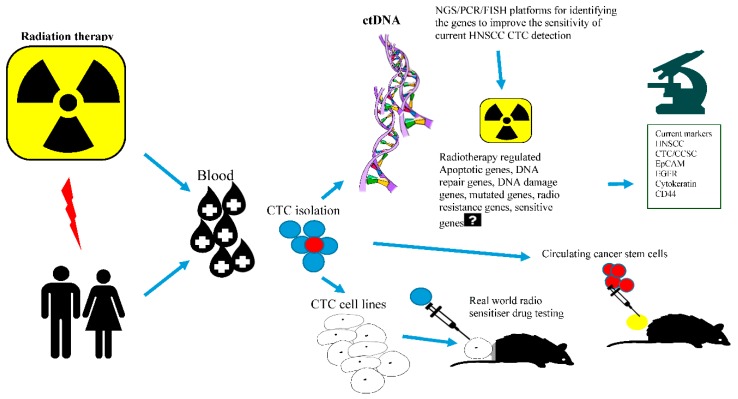
New developments and progress towards unexplored CTC-Cancer research. Figure generated using free smart draw templates and examples program.

**Table 1 cancers-11-00367-t001:** Overview of head and neck circulating tumour cells studies.

Author/Country	Head and Neck Cancer Types and Staging	CTC/CCSC Markers	Methodology	Patient Treatment
Kulasinghe, Zhou et al. Australia. [44]	Stage I–IV	CTC clusters, DAPI, cytokeratin EGFR, anti-CD45	Straight microfluidic chip	treatment naive
Kulasinghe, Kapeleris et al. Australia [45]	Stage I–IV	CTC clusters, PD-L1, ALK, EGFR, DAPI, anti-CD45	ClearCell FX system CTChip^®^ (Clearbridge Biomedics, Singapore)	treatment naive
Kulasinghe, Schmidt et al. Australia [46]	Stage I–IV	CTC clusters, DAPI, cytokeratin EGFR, anti-CD45	Spiral, microfluidic chip technology	treatment naive
Kawada, Takahashi et al. Japan [47]	TNM I–IV	anti-cytokeratin (CK) 8, 18 and 19, anti-epithelial cell adhesion molecule (EpCAM), and anti-CD45	cell sieve low pressure micro filtration assay	surgery and chemo radiotherapy
Fanelli, Oliveira et al. Brazil [48]	Locally advanced head and Neck Squamous cell carcinoma	CTM- TGF-βRI	ISET Method	adjuvant chemo radiotherapy, definitive radiotherapy (RT) concurrent with chemotherapy or cetuximab, or induction chemotherapy followed by RT concurrent with chemotherapy or cetuximab
Kulasinghe, Perry et al. Australia [49]	Supraglottal squamous cell carcinoma (SCC)	PD-1/PD-L1	CellSearch System (Janssen Diagnostics) and shear spiral microfluidic technology	single case study who do not respond to 1st/2nd line chemo-radiotherapy
Kulasinghe, Tran et al. Australia [3]	T1N0M0, T2/T3 (71%) and either no nodal spread N0 (50 (T4N2b))	CTC clusters, ICC and DNA FISH for EGFR DAPI Cytokeratin	shear spiral microfluidic technology	chemotherapy lung metastasis findings
Kulasinghe, Kenny et al. Australia [11]	T4 (55.1%) to advanced nodal spread (N2A-C) (62.1%).	CD45 APC, DAPI, CK-PE	CellSearch^®^ platform, ScreenCell^®^ (microfiltration device, France and RosetteSep™ (negative enrichment))	chemotherapy docetaxel, cisplatin and 5-fluorouracil
Kulasinghe, Perry et al. Australia [50]	Advanced HNC clinical stage patients ranging from T3N0 to T4aN2b.	Cytokeratin 8, 18, 19, CD45 and DAPI	RosetteSep^TM^ Human CD45 depletion cocktail (Stemcell Technologies™, Vancouver, BC, Canada), CTC culture for therapeutic screening	showed high CTC counts than EpCAM Cellsearch^®^ in HPV + patients
Patel, Shah et al. India [29]	Oral squamous cell carcinoma (OSCC)	CCSC CD44v6 and Nanog	CD44+ FITC labelled antibody by immuno-magnetic cell separation technique	CCSC resistance to Cisplatin
Morosin, Ashford et al. Australia [51]	Metastatic cutaneous squamous cell carcinoma affecting the lymph nodes of the parotid and/or neck	EpCAM and cytokeratin	Ficoll-Paque PLUS (GE Healthcare, NSW, Australia), IsoFluxTM immunomagnetic beads with Anti-EpCAM antibodies; CTC Enrichment kit Fluxion Biosciences Inc. System, Alameda, CA, USA	surgical treatment
Wu, Mastronicola et al. France [52]	Oral squamous cell carcinoma T4N2M0	EGFR neg CK PE-cytokeratin phycoerythrin DAPI CD45 APC EGFR	CellSearch^®^ system	rare case study pre-operative: intra-operative: and post-operative
Li, Liu et al. China [53]	Nasopharyngeal carcinoma (NPC) early TNM stages, Tumour stage I–IV	CD45 DAPI	Anti CD45 antibody conjugated immuno-magnetic beads (Cyttel, Jiangsu, China)	chemo radiotherapy treatment
Hsieh, Lin et al. Taiwan [54]	Locally advanced, recurrent, or initially metastatic HNSCC TNM II–IV	EpCAM/ podoplanin-PDPN	Power Mag immunofluorescence	Prechemotherapy Healthy donors
Inhestern, Oertel et al. Germany [31]	TNM III–IV	EpCAM	Laser scanning cytometry	PB/before/during/surgery radiotherapy, chemo radiotherapy
Grisanti, Almici et al. Italy [55]	TNM III–IV	EpCAM/CD45/DAPI	CellSearch	PB/before/during
Weller, Nel et al. Germany [56]	TNM I–IV	CK/CD45/DAPI	Immunofluorescence	PB/before treatment
Tinhofer, Konschak et al. Germany [57]	TNM I–IV	EGFR mRNA	RT-PCR	PB/before treatment
Gröbe, Blessmann et al. Germany [58]	TNM I–IV	EpCAM/CD45/DAPI	CellSearch	PB/before treatment
Bozec, Ilie et al. France [59]	TNM III–IV	EpCAM/CD45/DAPI	CellSearch	PB/before treatment
He, Li et al. China [60]	TNM III–IV	EpCAM/CD45/DAPI	CellSearch	PB/before treatment
Buglione, Grisanti et al. Italy [61]	TNM I–IV	EpCAM/CD45/DAPI	CellSearch	PB/before treatment
Nichols, Lowes et al. Britain [62]	TNM III–IV	EpCAM/CD45/DAPI	CellSearch	PB/before treatment
Balasubramanian, Lang et al. USA [63]	Head and neck cancer patients	EGFR cytokeratin, DAPI	NH_4_Cl and CD45 negative Immunomagnetic separation	surgical resection
Hristozova, Konschak et al. Germany [64]	TNM I–IV	EpCAM, CK	Flow cytometry	PB/before treatment
Jatana, Balasubramanian et al. USA [65]	TNM I–IV	CK, CD45, DAPI	Immunocytochemistry	PB/before treatment
Toyoshima, Vairaktaris et al. Germany [66]	TNM I–IV	CK20	RT-PCR	PB/after treatment
Mollaoglu, Vairaktaris et al. Greece [67]	Oral squamous cell carcinoma (OSCC) TNM I–IV, for the early detection of metastatic cells	Melanoma Associated Antigen- (MAGE A1, 2, 4 and 12)	disseminated tumour cell detection by bDNA technology,	Before treatment had floor of the mouth tumors
Yang, Lang et al. USA [68]	squamous cell carcinoma of the oral cavity, oropharynx, hypopharynx or larynx	EGFR, DAPI	NH_4_Cl and CD45 negative Immunomagnetic separation	surgical resection for patients who have not been previously treated for this disease.
Balasubramanian, Yang et al. USA [69]	Head and neck cancer patients	8, 18, 19 cytokeratins CD45, CD44, EpCAM Vimentin N-cadherin	NH_4_Cl and CD45 negative Immunomagnetic separation	surgical resection
Winter, Stephenson et al. Australia [70]	Advanced head and neck cancers	ELF3, CK19, EGFR and EphB4.	Immunomagnetic enrichment, RT-PCR	time of surgery
Guney, Yoldas et al. Turkey [71]	TNM I–IV	EpCAM	magnetic cell separation MACS	PB/before treatment
Wollenberg, Walz et al. Germany [72]	TNM I–IV	CK 19	IHC- alkaline phosphatase-anti-alkaline phosphatase (APAAP)	BM/before treatment
Wirtschafter, Benninger et al. USA [73]	TNM I–IV	CK 20	Immunocytochemistry	PB/before treatment

EpCAM-Epithelial cell adhesion molecule, CK-cytokeratin, Immunohistochemistry, NH_4_Cl-Ammonium Chloride, PB-Peripheral blood, BM-bone marrow, CD-cluster of differentiation, RT-PCR, reverse transcription–polymerase chain reaction, TNM-Tumour nodal metastasis.

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
