# Peer review of "Circulating Tumour Cells (CTC), Head and Neck Cancer and Radiotherapy; Future Perspectives"

_cancers, 2019, doi:10.3390/cancers11030367_

Round 1
Reviewer 1 Report
The manuscript (Cancers; DRUDIS_2017_68), entitled " Circulating Tumour Cells (CTC), Head and Neck 2 Cancer and Radiotherapy; Future perspectives" by Vanathi Perumal, and his colleagues describe the followings:
Head and neck cancer is the seventh most common cancer in Australia and globally.
Inspite of the current improved treatment modalities, there is still up to 50-60% local regional recurrence and or distant metastasis.
High-resolution medical imaging technologies such as PET/CT 22 and MRI do not currently detect the early spread of tumour cells, thus limiting the potential for an effective minimal residual detection and early diagnosis.
Circulating tumour cells (CTCs) are a rare subset of cells that escape from the primary tumour and enter into the bloodstream to form metastatic deposit or even re-establish themselves in the primary site of the cancer.
These cells are more aggressive and accumulate gene alterations by somatic mutations that are the same or even higher than the primary tumour because of additional features acquired in the circulation.
The potential application of CTC for clinical use is to acquire a liquid biopsy, by taking a reliable minimally invasive venous blood sample, for cell genotyping during radiotherapy treatment to monitor the decline in CTC detectability, mutational changes in response to radiation resistance and radiation sensitivity.
Currently, very little has been published on radiation therapy, CTC, and circulating cancer stem cells (CCSCs).
The prognostic value of CTC in cancer management and personalised medicine for head and neck cancer radiotherapy patients requires a deeper understanding at the cellular level along with other advanced technologies.
With this goal, this review summarises the current research of head and neck cancer CTC, CCSC and the molecular targets for personalised radiotherapy response.
Based on these studies authors conclude that CTC plays a major role in head and neck cancer and its therapy. This is a well-done study with proper rationale, and results do support the conclusion. The manuscript is well written.
Specific comments:
Maybe all the targeted therapies with overall survival data for HNSCC can be tabulated.
Maybe cost of treatment can also be highlighted as a separate paragraph.
Author Response
All four reviewer's response attached as one PDF file -cover letter by the corresponding author.

Reviewer 2 Report
The authors have done a good job in making the manuscript more clear and focused. This should be of interest to the readership.
Author Response
All four reviewers’ response attached as one PDF file -cover letter by the corresponding author

Reviewer 3 Report
Dear authors: Thank you for responding my suggestion, The manuscript is now better organised and more understandable, all of my suggestions were responded adequately. Thus, I recommend this manuscript for publication.
Author Response

(The authors gave the same response as above.)

Reviewer 4 Report
This review is a cluster of previous reports. Although a lot of information is included, they are fragmented and not organized well for readers' understanding. Even Table 1 is a enumeration of the references, and readers can not understand which is cutting-edge and important. Where are arrows in Figure 1 or 2? Less consistency is found through the paper.
Author Response

(The authors gave the same response as above.)

Round 2
Reviewer 4 Report
The authors revised the manuscript a couple of times, resulting in the improvement of the quality. Consistency are kept through the paper. Only a few points should be reconsidered.
Line 67, CSCs needs explanation of cancer stem cells. It is written in line 215 (too late).
Line 135, (hTERT) activity specificto should be specific to
Line 138-141, hyperbaric oxygen therapy , low dose cisplatin, or high dose cisplatin. How are they associated with CTC?
Line 224, Wnt pathway antagonist (sFRP4) in HNSCC CSCs decreased CD44 stem cell
markers and ALDH, ABCG2 and ABCC4 drug-resistance expression markers." and " is repeated. It should be grammatically corrected.
Line 242, "only CSCs are able to give rise to metastases." It is still a hypothesis in HNSCC.
Line 215, The major reason for treatment failure in anticancer therapy modalities is due to presence of CSCs, which are more resistant and highly tumorigenic with self-renewing capacity compared to CTCs expressing stem cell markers are known as circulating cancer stem cells (CCSCs). Is this sentence grammatically correct?
Author Response
The two points that have been of ongoing concern have now been addressed. Minor changes highlighted in green.
